# Exercise Training-Enhanced Lipolytic Potency to Catecholamine Depends on the Time of the Day

**DOI:** 10.3390/ijms21186920

**Published:** 2020-09-21

**Authors:** Hisashi Kato, Junetsu Ogasawara, Hisashi Takakura, Ken Shirato, Takuya Sakurai, Takako Kizaki, Tetsuya Izawa

**Affiliations:** 1Organization for Research Initiatives and Development, Doshisha University, 1-3 Tatara-Miyakodani, Kyotanabe City, Kyoto 610-0394, Japan; hkato@mail.doshisha.ac.jp; 2Faculty of Health and Sports Science, Doshisha University, 1-3 Tatara-Miyakodani, Kyotanabe City, Kyoto 610-0394, Japan; htakakur@mail.doshisha.ac.jp; 3Department of Health Science, Asahikawa Medical University, 2-1-1-1 Midorigaoka-Higashi, Asahikawa, Hokkaido 078-8510, Japan; junetsu@asahikawa-med.ac.jp; 4Department of Molecular Predictive Medicine and Sport Science, Kyorin University of School Medicine, 6-20-2 Shinkawa, Mitaka, Tokyo 181-8611, Japan; shirato@ks.kyorin-u.ac.jp (K.S.); sakutaku@ks.kyorin-u.ac.jp (T.S.); kizaki@ks.kyorin-u.ac.jp (T.K.); 5Graduate School of Health and Sports Science, Doshisha University, 1-3 Tatara-Miyakodani, Kyotanabe City, Kyoto 610-0394, Japan

**Keywords:** exercise training, timing of exercise, adipocyte, lipolysis, circadian rhythm

## Abstract

Exercise training is well known to enhance adipocyte lipolysis in response to hormone challenge. However, the existence of a relationship between the timing of exercise training and its effect on adipocyte lipolysis is unknown. To clarify this issue, Wistar rats were run on a treadmill for 9 weeks in either the early part (E-EX) or late part of the active phase (L-EX). L-EX rats exhibited greater isoproterenol-stimulated lipolysis expressed as fold induction over basal lipolysis, with greater protein expression levels of hormone-sensitive lipase (HSL) phosphorylated at Ser 660 compared to E-EX rats. Furthermore, we discovered that Brain and muscle Arnt-like (BMAL)1 protein can associate directly with several protein kinase A (PKA) regulatory units (RIα, RIβ, and RIIβ) of protein kinase, its anchoring protein (AKAP)150, and HSL, and that the association of BMAL1 with the regulatory subunits of PKA, AKAP150, and HSL was greater in L-EX than in E-EX rats. In contrast, comparison between E-EX and their counterpart sedentary control rats showed a greater co-immunoprecipitation only between BMAL1 and ATGL. Thus, both E-EX and L-EX showed an enhanced lipolytic response to isoproterenol, but the mechanisms underlying exercise training-enhanced lipolytic response to isoproterenol were different in each group.

## 1. Introduction

It is widely accepted that daily exercise training prevents central adiposity, obesity, and obesity-related disorders, and improves their consequences. Exercise training-induced improvements in adiposity and obesity-related disorder indices have been suggested to involve enhanced adipocyte lipolysis and amelioration of both adipokine dysregulation and adipose tissue inflammation [1,2,3,4,5]. Among these effects, exercise training-enhanced adipocyte lipolysis in response to hormone challenge [2,6,7,8,9,10,11,12,13,14] manifests in the loss of absolute lipid content in white adipocytes through intracellular lipid hydrolysis. Thus, it is a highly effective tool for positively regulating excessive energy expense.

Hormonal activation of adipocyte lipolysis is mediated by a cyclic adenosine monophosphate (cAMP)-dependent process comprising orchestrated activation of several lipolytic machineries, such as cAMP-dependent protein kinase (PKA), its anchor protein (AKAP) 150 [10], adipose tissue triacylglycerol lipase (ATGL) [15,16,17], perilipin [18,19], comparative gene identification-58 (CGI-58) [20], and hormone-sensitive lipase (HSL) [10,15,16,17,18,19,20,21]. Of these, HSL and ATGL expression depends on a transcriptional auto-regulatory feedback loop consisting of the core clock genes *Bmal1* and *Per2*: The dimers of CLOCK: BMAL1 directly modulate the expression of *Hsl* and *Atgl* mRNA by binding their E-box sequences, thereby resulting in circadian variation of lipolysis [22,23,24]. Therefore, it is hypothesized that exercise training at different times of the day with differing *Bmal1*/*Per2* expression can induce different lipolytic responses of adipocytes to hormone challenge. Exercise training-enhanced adipocyte lipolysis is well known to be closely associated with protein expression, localization, and phosphorylation of HSL and/or ATGL [2,6,7,8,10,14]. However, no direct evidence for this relationship has been reported to date. 

The present experiment was designed to verify this hypothesis by comparing the effects of two types of exercise training performed at different times of the day, on adipocyte lipolysis in response to catecholamine.

## 2. Results

### 2.1. Effect of Clock Gene mRNA Expression on Adipocyte Lipolysis

Before exploring the relationship between exercise timing and adipocyte lipolysis in response to hormone challenge, we first verified the relationship between *Bmal1* mRNA expression and isoproterenol-stimulated adipocyte lipolysis. As shown in Figure 1, following serum shock, temporal variations in *Bmal1* and *Per2* mRNA expression were observed in differentiated 3T3-L1 adipocytes (Figure 1A–C). These results are in agreement with previous reports [25,26,27]. *Bmal1* expression showed a clear circadian rhythm until time 60 h (ZT60) with two troughs at approximately ZT24 and ZT48, and two peaks at approximately ZT36 and ZT60. However, no clear circadian variation was detected after ZT60. In contrast, *Per2* expression troughs were found at ZT12 and ZT36, whereas its peaks were found at ZT24 and ZT48. These oscillations disappeared at 48 h after serum shock.

Next, we determined the associations between clock gene mRNA expression levels and adipocyte lipolysis. As shown in Figure 1D, isoproterenol-stimulated glycerol release, but not basal glycerol release, was significantly greater at ZT36, i.e., the time point of peak *Bmal1* expression, than that at ZT24, which was the trough of *Bmal1* mRNA expression. The levels of HSL phosphorylated at Ser563 and at Ser660 were also significantly greater at ZT36 than at ZT24 (Figure 1E–G).

### 2.2. Effect of *Bmal1* Knockdown on Adipocyte Lipolysis

To further obtain evidence for the effect of the circadian clock on adipocyte lipolysis, we knocked down BMAL1 in 3T3-L1 adipocytes. At 48 h following small interfering RNA (siRNA) transfection, *Bmal1* mRNA and protein expression was significantly decreased in siRNA-treated 3T3-L1 adipocytes compared to that in vehicle siRNA-treated cells (Figure 2A,B). In accordance with the changes in *Bmal1* levels, the mRNA expression of *Hsl* and *Atgl*, and protein expression levels of HSL, ATGL, and Perilipin1 were significantly lower in the *Bmal1* siRNA group compared with those in the vesicle siRNA group (Figure 2C,D). In accordance with the reduced levels of *Bmal1*, the mRNA and protein expression levels of HSL and ATGL were significantly lower in the *Bmal1* siRNA group compared with those in the control siRNA group (Figure 2C,D). In addition, the expression levels of Perilipin1, a scaffold protein of HSL on the surface of lipid droplets, was also significantly reduced in the *Bmal1* siRNA group (Figure 2D). The expression levels of protein kinase A (PKA)-anchoring protein 150 (AKAP150) protein, which promotes the binding of PKA and its substrate, and the regulatory subunit RIα of PKA were significantly decreased in the *Bmal1* siRNA group compared with those in the control siRNA group, but there were no changes in the levels of the regulatory subunits RIIα and RIIβ of PKA proteins (Figure 2E–G). Moreover, isoproterenol-induced but not basal adipocyte lipolysis was significantly blunted in BMAL1 knockdown 3T3-L1 adipocytes, as the levels of Ser563- and Ser660-phosphorylated HSL were significantly lower in the *Bmal1* siRNA group compared to those in the vesicle siRNA group (Figure 2H–K). Together, these results clearly indicate that the circadian clock might play an important role in adipocyte lipolysis in response to isoproterenol, and strongly confirm the findings of previous reports [22,23,24,25,26,27].

### 2.3. Circadian Oscillations in Bmal1 and Per2 Expression Patterns in Epididymal Adipose Tissue, and Physical Characteristics of the Rats

Based on the above findings, we verified the circadian clockwork oscillation in epididymal adipose tissue, and determined the exercise timing for chronic exercise training. As shown in Figure 3A, *Bmal1* and Per2 mRNA levels exhibited clearly opposite circadian oscillations in rat epididymal adipose tissue. *Bmal1* mRNA gradually rose during the dark period and reached a peak at Zeitgeber time (ZT) = 0; thereafter, it decreased rapidly. In contrast, *Per2* mRNA expression exhibited the opposite pattern. These results indicate that the circadian clockwork can oscillate accurately in adipose tissue similar to that described in previous studies [22,23,24,25,26,27]. Consequently, because *Bmal1* mRNA expression in epididymal adipose tissue peaked at ZT0 (08:00) and reached a trough at ZT12 (20:00), we established training groups for which the exercise timing coincided with these separate points: The rats that performed exercise training in the late part of the active phase (L-EX) ran on a treadmill at ZT = 22 (06:00), corresponding to the time point of the late part of the active phase with higher *Bmal1* expression; the rats that performed exercise training in the early part of the active phase (E-EX) ran at ZT = 12 (20:00), corresponding to the time point of the early part of the active phase with lower *Bmal1* expression (Figure 1). Corresponding sedentary control groups were designated as L-SED and E-SED, respectively. Thus, we selected two time points to explore whether the potency of exercise training-enhanced adipocyte lipolysis might be altered depending on the time of day of the exercise session.

Body weight gain in E-EX and L-EX rats was significantly blunted from the 6-week time point of exercise training onward, compared with that in E-SED and L-SED rats, respectively (Figure 3B). The final body weight was markedly lower in L-EX rats than in E-EX rats at the 9-week time point of exercise training (Table C in Figure 3). To verify the effect of exercise training at different times of the day on adipose tissue growth, the masses of several adipose depots were measured. The masses of epididymal, mesenteric, and inguinal adipose tissues were also significantly lower in L-EX than in E-EX rats; however, retroperitoneal adipose tissue weight did not differ significantly between these groups (Table C in Figure 3). L-EX rats also showed less food intake than the other groups. Overall, these results suggested that the time of day of the exercise significantly affected adipose tissue growth.

### 2.4. Adipocyte Lipolysis

As it is well known that exercise training enhances adipocyte lipolysis in epididymal adipose tissue [6,7,8,9,10,11,12,13,14], we also focused on the effect of exercise training on adipocyte lipolysis in this fat depot. In accordance with previous studies, isoproterenol-stimulated adipocyte lipolysis was significantly greater in exercised groups than in each respective control group (Figure 4A). In sedentary control rats, neither basal nor isoproterenol-stimulated adipocyte lipolysis differed significantly between L-SED and E-SED groups. Isoproterenol-stimulated adipocyte lipolysis expressed as glycerol release per mg of adipocytes tended to be greater in L-EX than in E-EX rats, although this did not reach statistical significance (Figure 4A). On the contrary, isoproterenol-stimulated lipolysis expressed as fold induction over basal lipolysis was significantly greater in the L-EX group than that in E-EX animals (Figure 4B). Under these conditions, the levels of both Ser563- and Ser660-phosphorylated HSL were significantly higher in isoproterenol-stimulated adipocytes than in adipocytes under basal status in all groups (Figure 4D,F), and those expressed as fold change over each respective basal level were significantly greater in the L-SED/EX groups than those in the E-SED/EX groups (Figure 4D,F). Compared with E-EX rats, L-EX rats showed greater isoproterenol-stimulated Ser660-phosphorylation of HSL expressed as fold change over the basal level (Figure 4G).

### 2.5. Changes in Lipolytic Machineries

We next examined the protein expressions of other lipolytic machineries. Hormone-stimulated adipocyte lipolysis is controlled by a lipolytic cascade consisting of several components, with HSL–perilipin1 and ATGL–CGI-58 complexes known to play pivotal roles in the process [15,16,17,18,19,20,21]. The expression levels of total ATGL, perilipin1, and CGI-58 proteins did not differ between the two sedentary control groups (Figure 5A,C–E), whereas the expression level of peroxisome proliferator-activated receptor (PPAR)γ2 tended to be lower (*p* = 0.06), and that of HSL was significantly lower, respectively, in L-SED rats compared to that in E-SED rats (Figure 5B,F). In the exercising groups, E-EX rats exhibited greater expression of ATGL protein but not of other proteins compared to E-SED rats, whereas L-EX rats showed greater expression levels of HSL, ATGL, perilipin1, and PPARγ2 compared to L-SED rats (Figure 5B–F). Consequently, L-EX rats showed greater increases in HSL, ATGL, perilipin1, and PPARγ2 protein levels as fold change over L-SED rats, whereas E-EX rats exhibited greater increases in only the ATGL protein level as fold change over E-SED rats (Figure 5G). Furthermore, we examined the level of AKAP150 and the regulatory subunits (RIα, RIIα, and RIIβ) of PKA. In the exercising groups, E-EX rats did not showed changed expression levels of all proteins (AKAP150, PKA–RIα, –RIIα, and –RIIβ), whereas L-EX rats exhibited greater expression levels of AKAP150, PKA–RIα, –RIIα, and –RIIβ proteins compared to L-SED rats (Figure 5H–L). Figure 5M shows the expression level of *Bmal1* mRNA in the samples used in the in vitro experiments. The expression level of *Bmal1* mRNA was considerably higher in L-SED and L-EX than that in E-SED and E-EX. However, its levels were comparable in E-EX and E-SED, and in L-EX and L-SED. These results suggest that exercise training performed in the late part of the active phase could enhance the expression of lipolytic machineries.

### 2.6. Association of BMAL1 Protein with Lipolytic Machineries

Finally, we verified the possible association of BMAL1 protein with HSL and ATGL, and, if any, the effects of exercise regimens at different time points on its associations were also determined. Furthermore, possible associations of BMAL1 with AKAP150 and the regulatory subunits of PKA were determined, because exercise training enhances the expression of AKAP150 and the regulatory subunits of PKA [10], which are compartmentalized within adipocytes and stimulate the translocation and activation of HSL. As shown in Figure 6, immunoprecipitates of HSL, Perilpin1, AKAP150, and several regulatory subunits (RIα, RIIα, and RIIβ) of PKA with antibodies to BMAL1 revealed evidence of BMAL1 association with each protein. Furthermore, the immunoprecipitates of HSL, Perilipin1, AKAP150, and RIIβ of PKA were significantly greater in the L-EX rats than those in both E-EX and L-SED rats (Figure 6B,D,E,H). However, in E-EX rats, the immunoprecipitation of ATGL was significantly greater in E-EX rats than in E-SED rats (Figure 6C), whereas that of HSL (*p* = 0.08) and RIIβ of PKA (*p* = 0.07) tended to be lower in E-EX rats than in E-SED rats (Figure 6B,H).

## 3. Discussion

In the current study, we demonstrated that isoproterenol-stimulated lipolysis was greater in L-EX than that in E-EX, with greater expression of phosphorylated HSL at both Ser563 and Ser660. Moreover, L-EX rats showed a greater increase in the expression of HSL, Perilipin1, proliferator-activated receptor-γ (PPAR-γ)2, AKAP150, and the regulatory subunits of PKA proteins compared with E-EX rats. Under these conditions, we discovered that interaction of BMAL1 protein with HSL, Perilipin1, AKAP150, and the regulatory subunits of PKA was significantly greater in the L-EX group than that in E-EX group, indicating that exercise training performed in the late part of the active phase has a useful role in the adaptive change of lipolysis in adipocytes, which is upregulated by exercise training. To the best of our knowledge, the observed relationship between exercise timing and adipocyte lipolysis in response to hormone challenge is reported for the first time in this study.

Many physiological processes and behaviors exhibit prominent circadian variations over the course of the day. The circadian rhythms are generated by a transcriptional auto-regulatory feedback loop that involves the core clock genes *Bmal1* and *Per2* in most tissues, including various adipose depots in mammals [22,23,24]. We also found clear circadian oscillations in the *Bmal1* and *Per2* mRNA expression levels in the epididymal adipose tissues; oscillatory expression of circadian clock genes was also observed in differentiated 3T3-L1 adipocytes following serum shock. Importantly, BMAL1 knockdown in 3T3-L1 adipocytes decreased the expression of several lipolytic machineries and blunted isoproterenol-induced adipocyte lipolysis. These observations suggest that the lipolytic machineries and adipocyte lipolysis in response to isoproterenol are strongly associated with *Bmal1* and/or *Per2* expression.

However, even though *Bmal1* mRNA expression when the parameters were tested was significantly greater in the L-SED than in the E-SED rat, the expression level of total HSL protein was significantly lower in L-SED than in E-SED rats. This seems inconsistent with our findings and the previous understanding representing that the circadian variation of *Hsl* and *Atgl* mRNA expression is regulated by CLOCK/BMAL1 [22,23]. The observed phenomenon might be explained by a series of reactions induced by E-box stimulation of CLOCK/BMAL1 as described below. The expression rhythms of the two lipolytic genes, *Hsl* and *Atgl*, have been described to be in phase coherence with E-box-regulated genes, such as *Per2* [23]; in turn, PER2 can modulate the PPARγ axis in a posttranslational manner [24,28]. PPARγ2 subsequently upregulates the transcription of both HSL [29] and ATGL [6]. In the present study, the *Per2* mRNA level was lower in the time during ZT24 to ZT2, corresponding to the time of sample collection and parameters’ measurements in L-SED rats, than in the time during ZT14 to ZT16, corresponding to those in E-SED rats. Indeed, in synchronization with each phase of *Per2* mRNA, PPARγ2 protein expression tended to be lower in L-SED rats than in E-SED rats. This result is in good agreement with the previous finding that PPARγ2 exhibits a remarkable dual circadian expression pattern [26,30] showing a low expression level in the late part of the active phase but a high expression level during at least 8 h after the onset of the active period [26]. It is thus rational to consider that a lower tendency of PPARγ2 expression in L-SED rats compared with that in E-SED rats resulted in a lower expression of total HSL protein. Nevertheless, isoproterenol-stimulated lipolysis was not different between E-SED and L-SED rats. This result might be due to the greater fold-change of HSL phosphorylated at Ser563 and Ser660 over the basal condition in L-SED rats even if the expression level of total HSL was lower in L-SED rats than in E-SED rats.

According to the scenario described above, the PPARγ2 protein level in L-EX rats had to be lower than that of E-EX rats, but there was no significant difference in total HSL between E-EX and L-EX rats. This is considered to result from the lack of difference in the expression level of PPARγ2 levels between both groups, possibly because L-EX salvaged the time-of-day-dependent decline in PPARγ2 levels in the late part of the active phase. Thereby, the same levels of total HSL and ATGL were found in E-EX rats and L-EX rats. Indeed, exercise training has been described to robustly enhance the level [6] and transcriptional activation [31] of PPARγ2 in rodent white adipose tissue, though these studies do not provide information on the timing of the performed exercise training and the timing of collected parameters. Further, the effects of E-EX on PPARγ2 expression may have been masked by high levels of PPARγ2 lasting at least 8 h after the onset of the active period [28]. Under these conditions, L-EX rats compared with E-EX rats exhibited greater expression of lipolytic machineries expressed as fold change over the respective SED rats, resulting in an enhanced potency of their adipocytes for isoproterenol-induced lipolysis.

Our co-immunoprecipitation assays add a new notion to the underlying mechanism above: BMAL1 protein was associated with several lipolytic machineries tested, including HSL, ATGL, Perilipin1, the regulatory subunits (RIα, RIβ, and RIIβ) of PKA, and AKP150. Of these lipolytic machineries, AKAPs, the scaffolding proteins, mediate the association between β-adrenergic receptor and PKA in adipocytes [32,33] through their association with either the RIIα or RIIβ subunit of PKA [10,34,35]. The interaction of AKAP with PKA has been described to play an essential role as one of the elements involved in controlling HSL activity, and exercise training increased the protein expression of AKAP150 with enhanced activity and expression of several PKA subunits (catalytic, RIIα, and RIIβ) [10]. In particular, the RIIβ isoform is the most expressed in white adipose tissue in mice [36,37]. RIIβ knockout mice (RIIβ^−/−^) remain remarkably lean even when challenged with a high-fat diet [37], resulting in the loss of responsiveness to β-agonists [38]. Thus, RIIβ has been demonstrated as the key PKA regulatory subunit [36,37,38,39]. Moreover, previous studies have reported that perilipin1, the predominant perilipin isoform in adipocytes, is a key regulator of both basal and PKA-stimulated lipolysis [40], and that AKAPs are capable of forming a complex with perilipin1 and PKA on the surfaces of lipid droplets in adipocytes [41]. In the present study, L-EX rats exhibited greater protein expression only for AKAP150 and the regulatory subunits of PKA-RIIα compared to the E-EX rats, whereas greater associations of BMAL1 were found with all the lipolytic machineries tested except for ATGL. In addition, compared to all the other groups, L-EX rats exhibited greater associations of BMAL1 with HSL, Perilipin1, AKP150, and the RIIβ subunit of PKA. Accordingly, a mechanism underlying a greater lipolytic potency of adipocytes to isoproterenol in L-EX rats compared to E-EX rats would involve their high interaction of BMAL1 with AKAP-PKA-HSL-Perilipin1, the locus for essential events in the sequential activation of triacylglycerol hydrolysis. By contrast, in a comparison between E-EX and E-SED rats, a greater co-immunoprecipitation was found only between BMAL1 and ATGL. This result may imply that a higher isoproterenol-stimulated lipolysis in E-EX rats compared with E-SED rats might depend largely on the enhanced expression of ATGL and its greater interaction with BMAL1. Thus, the coupling efficiency between BMAL1 and lipolytic machineries was also different between E-EX rats and L-EX rats, and this difference may contribute to the difference in lipolytic potency to isoproterenol between both rats. However, this is only speculation at this point as the BMAL1-binding domain on the lipolytic machineries tested has not so far been defined to date, and our data did not prove a functional association of BMAL1 with the lipolytic machineries in controlling the acute lipolytic response to hormone challenge.

In the present study, total energy intake was less in L-EX rats than in E-EX rats, and the differences in total expenditure including spontaneous locomotor activity other than treadmill running were not controlled. In particular, the latter data could be required to determine any effects of exercise training per se on adipocyte functions. In fact, a difference in the energy balance could be associated with differences in body weight and adipose tissue weight between groups. However, regarding the differences in the potency of isoproterenol-stimulated adipocyte lipolysis between both groups, based on evidence from animal studies, it should be noted that exercise training-enhanced adipocyte lipolysis is a true metabolic adaptation and not secondary to reduced adipocyte size [11,12,13,14], which is associated with reduced weight of the adipose tissue. Therefore, we believe that even if a difference in energy expenditure other than treadmill running had been reflected in the difference in adipose tissue weight, it may not have a big impact on the difference in lipolytic response between the groups. Further, the possible effects of diurnal variations, including glucose and lipid metabolism [42], and of hormones, such as cortisol and insulin, that affect the lipolytic response [43,44,45,46] should be considered more importantly. Finally, other factors that impact local protein expression/activation in adipocytes and that determine blood glycerol concentration, such as insulin levels, adipogenic versus lipolytic mechanisms, vascular tone, adrenergic activity, and substrate availability, should also be considered more systematically.

In summary, both E-EX and L-EX enhanced the lipolytic response to isoproterenol. However, mechanisms underlying the EX-enhanced lipolytic response to isoproterenol were different in each group, and L-EX resulted in stronger lipolytic potency, compared to E-EX. Notably, this difference may include disparities in the association levels of BMAL1 with several lipolytic machineries. Thus, exercise training performed in the late part of the active phase may be more effective for a relatively stronger potency of the adipocyte lipolytic response to catecholamine, at least in part resulting in a lesser gain of adipose tissue and body weight during growth. Our present results might also be associated with the recent finding that exercise in the late dark phase can ameliorate diet-induced obesity by affecting energy homeostasis rather than glucose homeostasis [46,47] and through enhanced metabolic activity in the skeletal muscles [48,49]. The energy metabolism of skeletal muscles during acute exercise has also been shown to differ greatly depending on the time of the day for a single bout of exercise [48,49]. In further studies, to establish the roles of exercise training in ameliorating obesity, data derived from fat depots other than epididymal adipose tissue should be acquired using a similar strategy to that in the current study in obesity animals, along with establishing how exercise training modulates the interaction of BMAL1 with the tested lipolytic machineries.

## 4. Materials and Methods

### 4.1. Bmal1 and Per2 mRNA Circadian Rhythm in Rat Epididymal Adipose Tissue

Four-week-old male Wistar rats (*n* = 24, SLC, Shizuoka, Japan) were housed in a temperature-controlled room at 23 °C with a 12:12 h light-dark cycle. Food and water were available ad libitum. All rats were allowed seven days to acclimatize to their new environment prior to establishing a normal light/dark cycle: ZT0 (08:00) indicates lights on, and ZT12 (20:00) indicates lights off. To confirm circadian clock gene rhythmicity in adipose tissue, *Bmal1* and *Per2* expression levels were measured every 3 h over a 24-h period (eight time points) in epididymal adipose tissue collected from 5-week-old rats. The animals were anaesthetized with an intraperitoneal injection of pentobarbital sodium (5 mg/100 g body weight; Kyoritsu Seiyaku, Tokyo, Japan) and killed by exsanguination through the abdominal aorta at ZT0, 3, 6, 9, 12, 15, 18, and 21 (*n* = 3 each). Epididymal adipose tissue was rapidly removed, immediately frozen in liquid nitrogen, and stored at −80 °C until RNA preparation. All animal protocols were approved by the Animal Care Committee of Doshisha University (A15010, A19005).

### 4.2. Exercise Training Program and Sample Collection Times

Four-week-old male Wistar rats (*n* = 24, SLC, Shizuoka, Japan) were acclimatized, and then randomly divided into two exercise training and their respective control groups (*n* = 6 each). Because *Bmal1* mRNA expression epididymal adipose tissue peaked at ZT0 (08:00) and reached a trough at ZT12 (20:00), we established training groups in which the exercise timing coincided with these separate points: The rats of L-EX ran on a treadmill at ZT = 22 (06:00), corresponding to the time point of the late part of the active phase with higher *Bmal1* expression; the E-EX group ran at ZT = 12 (20:00), corresponding to the time point of the early part of the active phase with lower *Bmal1* expression (Figure 1). The corresponding sedentary control groups were designated as L-SED and E-SED, respectively.

Exercise comprised running on a treadmill set at a 5° incline, 5 days per week for 9 weeks, as described previously [6,7,9,10]. Briefly, the initial training intensity was 15 m/min for 30 min; thereafter, the running speed and duration were progressively increased until after 6 weeks, when the rats ran continuously at 30 m/min for 90 min. Both groups started with a warm-up at 10 m/min, and thereafter the pace and the time of continuous running were increased gradually from 15 to 30 m/min and 30 min to 90 min over 9 weeks, respectively (detailed in Appendix A). The intensity of this training protocol was estimated to be 60–70% VO_2_ max based on the literature [50,51,52], and our previous study confirmed that the same protocol of exercise training enhanced citrate-synthase activity [53] and adipocyte lipolysis in response to isoproterenol [6,7,9,10]. The sedentary control rats were not subjected to treadmill running. Rat body weights and food intake were monitored daily until the end of the study.

At the end of the study, to avoid any influence of a last bout of exercise, the exercised rats were euthanized under anesthesia at 48 h after the end of the last 90-min exercise session (at ZT24 and ZT14 in the L-EX and E-EX groups, respectively). Subsequently, epididymal adipose tissue was rapidly removed and enzymatically dispersed to obtain the primary adipocytes from all groups.

### 4.3. Preparation of Primary Adipocytes

Adipocytes were isolated as described previously [6,7,9,10]. Briefly, fat pads were minced with scissors and placed in plastic vials in buffer A (Krebs-Ringer bicarbonate solution buffered with 10 mM HEPES, pH 7.4 containing 5.5 mM glucose and 2% (*w*/*v*) fatty acid-free bovine serum albumin) with 200 nM adenosine and 1 mg/mL collagenase type 1 (Worthington Biochemical, Lakewood, NJ, USA). Collagenase digestion was performed at 37 °C in a water-bath shaker. After 45 min, the vial contents were immediately filtered through a mesh and centrifuged at 100× *g* for 10 min. The layer of floating cells was then washed thrice with buffer A. A portion of fresh adipocytes was used immediately for examining the lipolytic response to isoproterenol; the remaining adipocytes were used for either immunoblotting or quantitative real-time polymerase chain reaction (PCR). The above sampling adjustments for E-EX/SED and L-EX/SED were completed at ZT2 and ZT16, respectively.

### 4.4. Lipolysis Assay

Freshly isolated adipocytes were incubated in plastic vials in a total volume of 300 μL of buffer A containing 0.05 mg/mL adenosine deaminase (Sigma, St. Louis, MO, USA). After 2 min of pre-incubation, adipocytes were incubated for 30 min with or without 50 μM isoproterenol to investigate their lipolytic responses. Then, the cell-free incubation medium was removed and assayed for glycerol release as an index of lipolysis. Glycerol release was determined using an Adipolysis Assay Kit (Cayman, Ann Arbor, MI, USA) according to the manufacturer’s instruction. Lipolysis was expressed as nmol of glycerol/mg of total protein/hour, because we previously confirmed that the same protocol of exercise training as that used in this study did not affect the protein contents per unit of cells (g/10^5^ cells) [10]. The protein in adipocytes was extracted as follows: Adipocytes were washed twice with phosphate-buffered saline and homogenized in 20 mM HEPES pH 7.5, 1% NP-40, 0.1% sodium dodecyl sulphate (SDS), 0.5% deoxycholic acid, and 150 mM sodium chloride, supplemented with protease and phosphatase inhibitors (ATTO, Tokyo, Japan). Protein was measured using a commercially available kit (BCA protein assay kit, Funakoshi, Tokyo, Japan).

### 4.5. Gene Expression Analysis by Quantitative Real-Time PCR

Total RNA was extracted using ISOGEN II (Nippon gene, Tokyo, Japan). First-strand cDNA was synthesized using the PrimeScriptTM II first standard cDNA Synthesis Kit (Takara, Shiga, Japan), following the manufacturer’s protocol. For real-time PCR, RNA was reverse transcribed using KAPA SYBR^®^ FAST ABI Prism^®^ qPCR Master Mix (KAPA BIO, Wilmington, MA, USA), then amplified using the Applied Biosystems StepOne ^®^ Real-Time PCR System (Applied Biosystems, Waltham, MA, USA). The amplification protocol included initial denaturation for 10 min at 95 °C followed by 40 cycles of 15 s denaturation at 95 °C, 1 min annealing at 60 °C, and 1 min extension at 72 °C. Relative expression was normalized to that of 18S ribosomal RNA using the 2^−ΔΔCt^ method. Amplification of specific transcripts was confirmed via melting curves between 68 and 95 °C following PCR. Primers sequences are listed in Appendix A.

### 4.6. Immunoblotting Analysis and Immunoprecipitation

Fresh adipocytes were washed twice with phosphate-buffered saline and homogenized in EzRIPA lysis buffer (ATTO) or T-PER^@^ Tissue Protein Extraction Reagent (Invitrogen, Waltham, MA, USA) with protease and phosphatase inhibitors. The homogenate was incubated on ice for 15 min and centrifuged for 20 min at 14,000× *g* (4 °C). The supernatant was recovered and further cleared by centrifugation. Samples were frozen at −80 °C until analyzed. Samples did not contain significantly different amounts of total protein (data not shown). Therefore, identical volumes of each sample were mixed with Laemmli sample buffer and heated for 2 min at 95 °C. After separation by 6–12.5% SDS-polyacrylamide gel electrophoresis, proteins were transferred to polyvinylidene fluoride membranes (ATTO), blocked for 60 min with Tris-buffered saline (TBS) containing 0.1% Tween-20 (TBS-T) and 5% skim milk, or with Block Ace ^®^ Powder (DSP, Osaka, Japan) dissolved in purified water, and were then probed at 4 °C overnight in Can Get Signal^@^ Immunoreaction Enhancer Solution 1 (TOYOBO, Osaka, Japan) against total HSL (Abcam, Cambridge, UK), HSL phosphorylated at Ser-563 (Cell Signaling Technology [CST], Danvers, MA, USA), HSL phosphorylated at Ser-660 (CST), ATGL (CST), total perilipin 1 (Abcam), CGI-58 (Santa Cruz Biotechnology, Dallas, TX, USA), PPARγ2 (ThermoFisher Scientific, Waltham, MA, USA), BMAL1 (Abcam), AKAP150 (Alomone Labs, Jerusalem, Israel), PKARIα (BD biosciences, Franklin Lakes, NJ, USA), PKARIIα (Abcam), PKARIIβ (Abcam), and β-actin (Abcam). The membranes were then labelled for 60 min with anti-rabbit or anti-mouse immunoglobulin G (1:2500; GE Healthcare, Chalfont, UK). Bands were visualized using an ECL Prime system (GE Healthcare) and quantified on a ChemiDocTM MP system (Bio-Rad, Hercules, CA, USA). Protein abundance was normalized to that of β-actin.

Immunoprecipitation of the regulatory subunits of PKA (RIα, RIIα, RIIβ), AKAP150, HSL, and ATGL was also performed. The cell lysate containing 1 mg/mL protein was used for immunoprecipitation. The samples prepared as above were incubated with antibodies against BMAL1, and the immunocomplexes were recovered by adsorption to Dynabeads™ Protein G (ThermoFisher Scientific). The immunocomplexes then were subjected to SDS-PAGE. The resolved proteins were transferred to PVDF membranes (ATTO), and the resultant blots were stained with antibodies against each protein mentioned above. Thereafter, the bands were visualized as described above.

### 4.7. Cell Culture of 3T3-L1 Mouse Embryo Fibroblasts

The cell culture procedures were identical to those described in our previous report [54]. Briefly, 3T3-L1 preadipocytes were seeded at 7500 cells/cm^2^, grown until confluence in low glucose Dulbecco’s modified Eagle’s medium (DMEM) containing 10% fetal bovine serum (FBS), and maintained in the same medium for 2 days post-confluence at 37 °C in 95% air and 5% CO_2_. Differentiation was induced by treatment with 1 μM dexamethasone, 0.5 mM isobutylmethylxanthine, 1 μg/mL insulin, and 10% FBS for 48 h. Cells were cultured for six additional days in DMEM containing 1 μg/mL insulin and 10% FBS. The medium was refreshed every 2 days. With this regime, fully differentiated adipocytes were obtained by day 8. To synchronize the molecular circadian clock in differentiated adipocytes, cells were treated with 50% horse serum as described previously [25]. Briefly, after obtaining fully differentiated adipocytes, the medium was exchanged for serum-rich medium (DMEM supplemented with 50% horse serum). After 2 h, this medium was replaced with serum-free OPTI-MEM (Invitrogen) at time 0. Every 4 h from the initiation of serum shock (time 0 h), a portion of the cells was harvested for quantitative real-time PCR or immunoblotting. In the remaining cells, the lipolytic response to isoproterenol was measured at 24 and 36 h, during the period of lower or higher *Bmal1* expression, respectively. The cells were treated with maximally stimulating concentrations of isoproterenol (10 μM), as described previously [55,56].

### 4.8. Transfection of 3T3-L1 Cells with Small Interfering RNA (siRNA)

To gain insight into the mechanism by which BMAL1 regulates the lipolytic cascade in adipocytes, 30 nM of predesigned siRNA targeting *Bmal1* (MISSION siRNA, Sigma-Aldrich, St. Louis, MO, USA) was transfected into differentiated 3T3-L1 adipocytes using Deliver-X siRNA Transfection Reagent (Affymetrix, Santa Clara, CA, USA) according to the manufacturer’s instruction. MISSION siRNA Universal Negative Control (Sigma) was used for the control experiments. After 48 h, the lipolytic response was analyzed and the cells were harvested for quantitative real-time PCR or immunoblotting.

### 4.9. Statistical Analysis

All data represent the means ± standard error of the mean. The results of animal experiments were analyzed using the Student’s *t*-test, one-way analysis of variance (ANOVA), or two-way ANOVA, with the exercise training state and exercise training timing as fixed factors where appropriate. Where the main effects were considered significant, the Bonferroni post hoc test for multiple comparisons was conducted. Furthermore, the Student’s *t*-test was used to calculate the significance of inter-group differences in experiments in vitro. All analyses were performed using Microsoft Excel (Redmond, WA, USA). All statistical analyses were conducted with a significance level of α = 0.05 (*p* < 0.05).

## 5. Conclusions

Our data presents evidence that exercise training performed in the late part of the active phase was associated with a stronger lipolytic potency of adipocytes in response to isoproterenol, compared with that after training in the early part of the active phase. This stronger potency may be due to the greater expression of lipolytic machineries expressed as fold change over the respective sedentary rats compared to their counterparts, and was accompanied by high association of BMAL1 protein with several lipolytic machineries, such as AKAP150, the regulatory subunits of PKA, HSL, and Perilipin1 proteins.

## Figures and Tables

**Figure 1 ijms-21-06920-f001:**
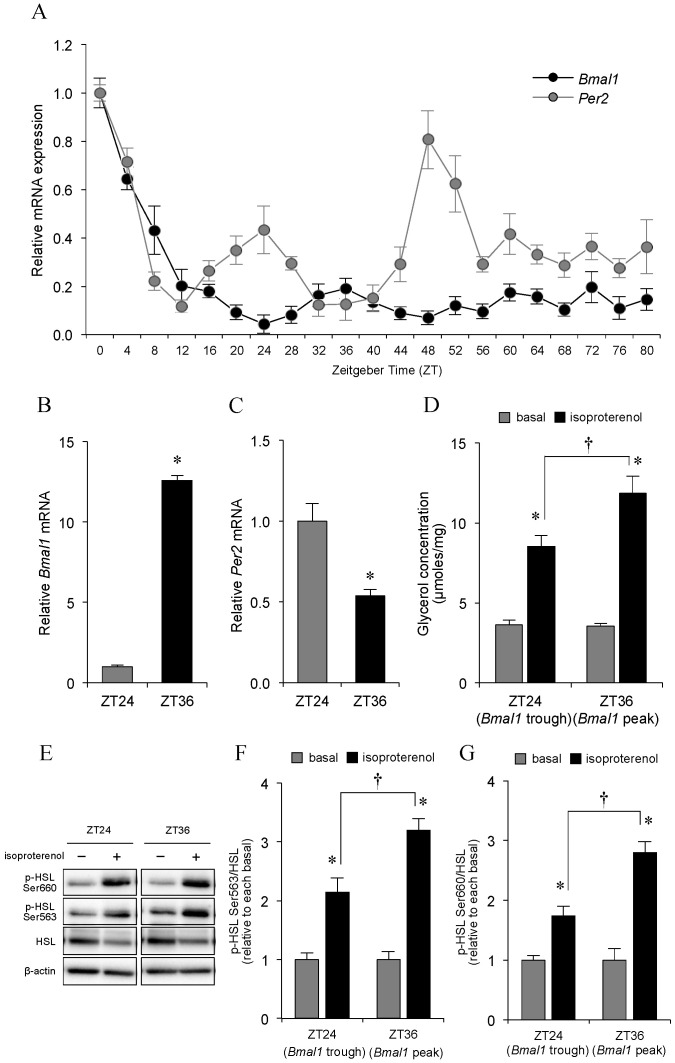
Effect of clock gene mRNA expression levels on adipocyte lipolysis in differentiated 3T3-L1 adipocytes. To induce circadian gene expression in differentiated 3T3-L1 adipocytes, cells were treated with 50% horse serum. After 2 h, the medium was replaced with serum-free OPTI-MEM medium and was harvested every 4 h from the start of the serum shock (Zeitgeber time (ZT) = 0). *Bmal1* and *Per2* mRNA expression levels were measured by quantitative real-time PCR using specific primers and were normalized to the level of 18S rRNA (Supplementary Data 1). (**A**) Changes in *Bmal1* and *Per2* mRNA, (**B**,**C**) Relative levels of *Bmal1* and *Per2* mRNA at ZT = 24 and ZT = 36 are illustrated with the mean value at ZT = 24 arbitrarily set to 1. (**D**) Comparison between adipocyte lipolysis at ZT24 and that at 36. At 1 h before ZT = 24 and ZT = 36, differentiated 3T3-L1 adipocytes were incubated with fresh medium with or without 10 μM isoproterenol. After 2 h, glycerol released into the medium was measured. (**E**) Representative expression levels of HSL phosphorylated at Ser563 and Ser660 were measured by Western blotting, and (**F**,**G**) normalized to total HSL levels (mean basal levels at ZT = 24 and ZT = 36 were arbitrarily set to 1). In all experiments, the values represent the means ± S.E. of three independent experiments. * *p* < 0.05, ZT24 vs. ZT36 or basal vs. isoproterenol, intergroup differences; ^†^
*p* < 0.05, isoproterenol stimulation at ZT24 vs. that at ZT36, intergroup differences.

**Figure 2 ijms-21-06920-f002:**
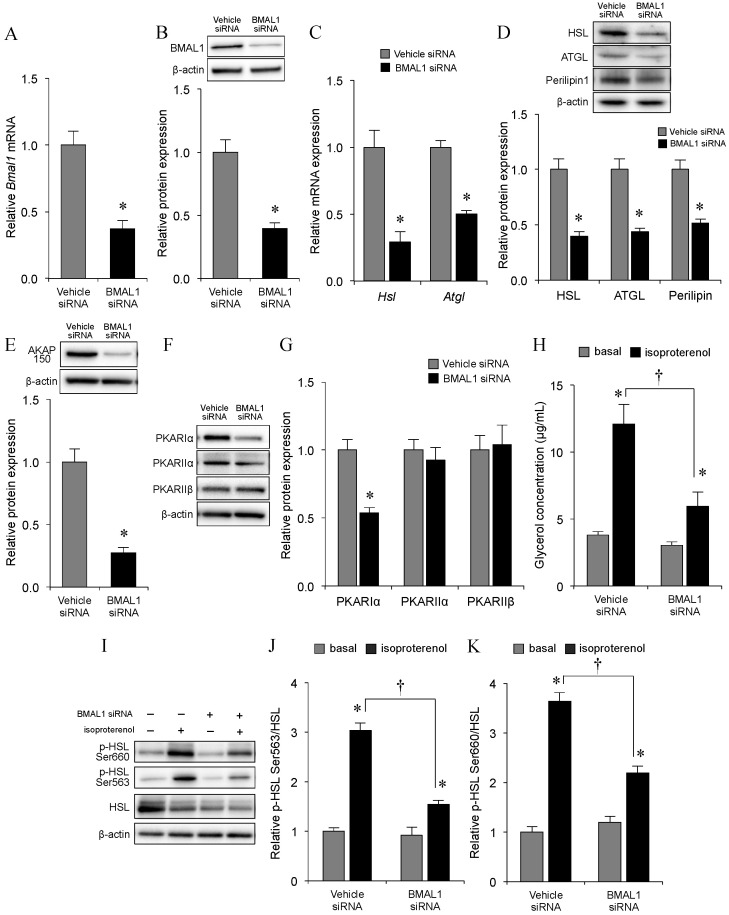
Effect of *Bmal1* expression knockdown on adipocyte lipolysis in differentiated 3T3-L1 adipocytes. Differentiated 3T3-L1 adipocytes were transfected with control siRNA or specific siRNA for *Bmal1* using a transfection reagent. After 48 h, basal or isoproterenol-stimulated glycerol release was measured, and the cells were harvested for subsequent analysis by quantitative real-time PCR or Western blotting. (**A**) Relative expression levels of *Bmal1* mRNA. The mRNA levels were measured by quantitative real-time PCR using specific primers and normalized to the amount of 18S rRNA. (**B**) Representative immunoblotting data (upper panel) and relative densities of BMAL1 protein (bottom panel) measured by Western blotting, normalized to β-actin. (**C**) Expression levels of *Hsl* and *ATGL* mRNA were measured by quantitative real-time PCR using specific primers and normalized to the amount of 18S rRNA. (**D**) Representative immunoblotting data (upper panel) and relative densities of the HSL, ATGL, and Perilipin1 proteins (lower panel) measured by Western blotting and normalized to β-actin. (**E**) Representative immunoblotting data (upper panel) and relative densities of AKAP150 protein (bottom panel) measured by Western blotting, and normalized to β-actin. (**F**) Representative immunoblotting data of PKA-RIα, -RIIα, -RIIβ, and β-actin proteins. (**G**) Relative densities of the PKA- RIα, -RIIα, and -RIIβ proteins measured by Western blotting, and normalized to β-actin. The values in the bar graphs in panels A, B, C, D, E, and G are shown relative to the mean levels detected after treatment with control siRNA, which was arbitrarily set to 1. (**H**) Reduced response of lipolysis to isoproterenol in *Bmal1*-siRNA adipocytes. (**I**) Representative immunoblotting data of HSL phosphorylated at Ser563 and Ser660. (**J**,**K**) Relative expression levels of HSL phosphorylated at Ser563 and Ser660 as measured by Western blotting and normalized to total HSL levels. The values in the bar graphs in panels H, J, and K are shown relative to the mean level detected in basal conditions, which was arbitrarily set to 1. In all experiments, values represent the means ± S.E. of three independent experiments. * *p* < 0.05, control-siRNA vs. *Bmal1*-siRNA or basal vs. isoproterenol; intergroup differences; ^†^
*p* < 0.05, isoproterenol stimulation of control siRNA vs. that of *Bmal1* siRNA.

**Figure 3 ijms-21-06920-f003:**
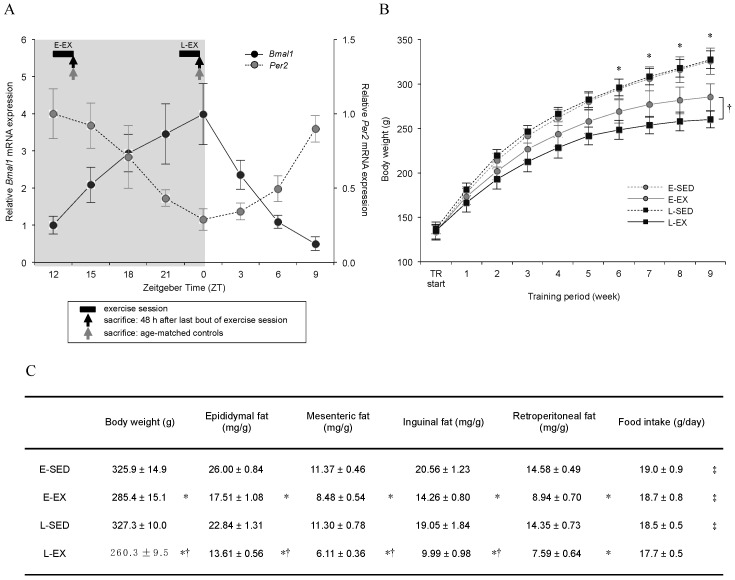
(**A**) Circadian oscillations in *Bmal1* and *Per2* expression patterns in epididymal adipose tissue and the sampling time. Epididymal adipose tissue was harvested from 5-week-old Wistar rats every 3 h over a 24-h period (eight time points, *n* = 3 in each time point). Total RNA was extracted from the collected epididymal adipose tissue and the expression levels of *Bmal1* and *Per2* mRNA were measured by quantitative real-time PCR and normalized to 18S rRNA levels. The values are shown as the means ± S.E. (**B**,**C**) Effect of exercise training timing on several physical parameters and food intake. The panel B shows changes in body weight during the study period, and table C shows the body weight and the weight of several fat pads at the end of the study, and the mean food intake during the study period. Body weights were measured every week in the morning. Exercise training-induced slowing of body weight gain was observed after six weeks of training time in both training groups compared to that in the control groups (*n* = 6 per group): E-EX, exercise in the early part of the active phase; L-EX, exercise performed in the late part of the active phase; E-SED, sedentary control for E-EX; L-SED, sedentary control for L-EX. The values are shown as the means ± S.E. * *p* < 0.05, SED vs. EX; ^†^
*p* < 0.05, E-EX vs. L-EX; ^‡^
*p* < 0.05, vs. L-EX.

**Figure 4 ijms-21-06920-f004:**
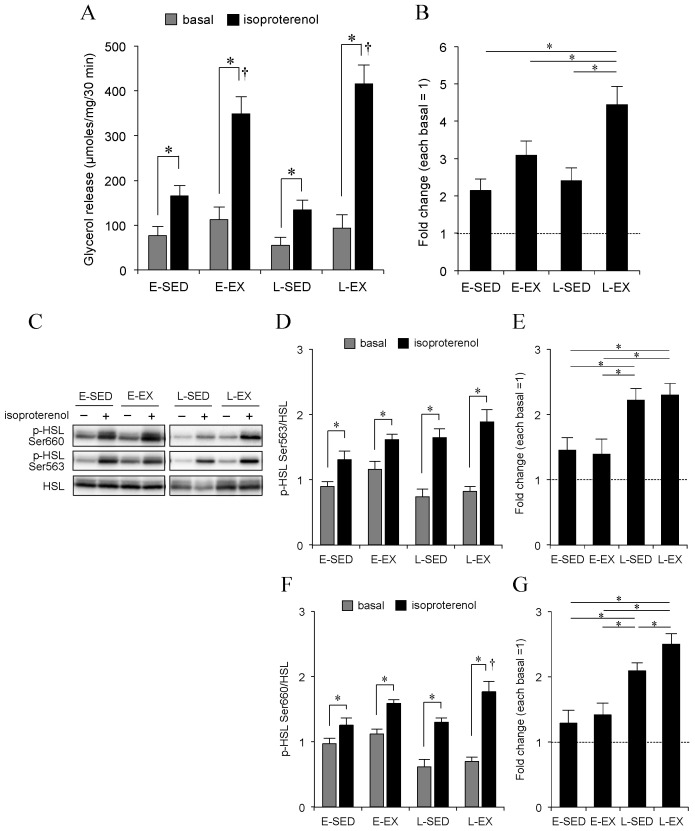
Effect of exercise training timing on lipolysis in basal conditions and after stimulation by isoproterenol in primary adipocytes isolated from epididymal adipose tissues. (**A**) Levels of basal and isoproterenol-induced glycerol release, as an index of lipolysis, from adipocytes isolated from the four groups: E-SED, L-SED, E-EX, and L-EX rats (*n* = 6 per group). (**B**) Isoproterenol-stimulated lipolysis expressed as fold induction over basal lipolysis in the groups (levels in respective sedentary controls were set to 1). (**C**) Representative expression of HSL phosphorylated at Ser563 and Ser660 as measured by Western blotting. (**D**) Relative expression levels of HSL phosphorylated at Ser563 to the total HSL level, and (**E**) its expression over respective basal conditions. (**F**) Relative expression levels of HSL phosphorylated at Ser660 to total HSL level, and (**G**) its expression over respective basal conditions. The values of the bar graphs in panels B, E, and G are shown relative to the mean level detected in basal conditions, which was arbitrarily set to 1. In all experiments, values represent the means ± S.E. from 6 rats in each group. * *p* < 0.05, or less. ^†^
*p* < 0.05 or less, E-SED vs. E-EX or L-SED vs. L-EX. The abbreviations for the groups of rats are shown in Figure 2 and Figure 3.

**Figure 5 ijms-21-06920-f005:**
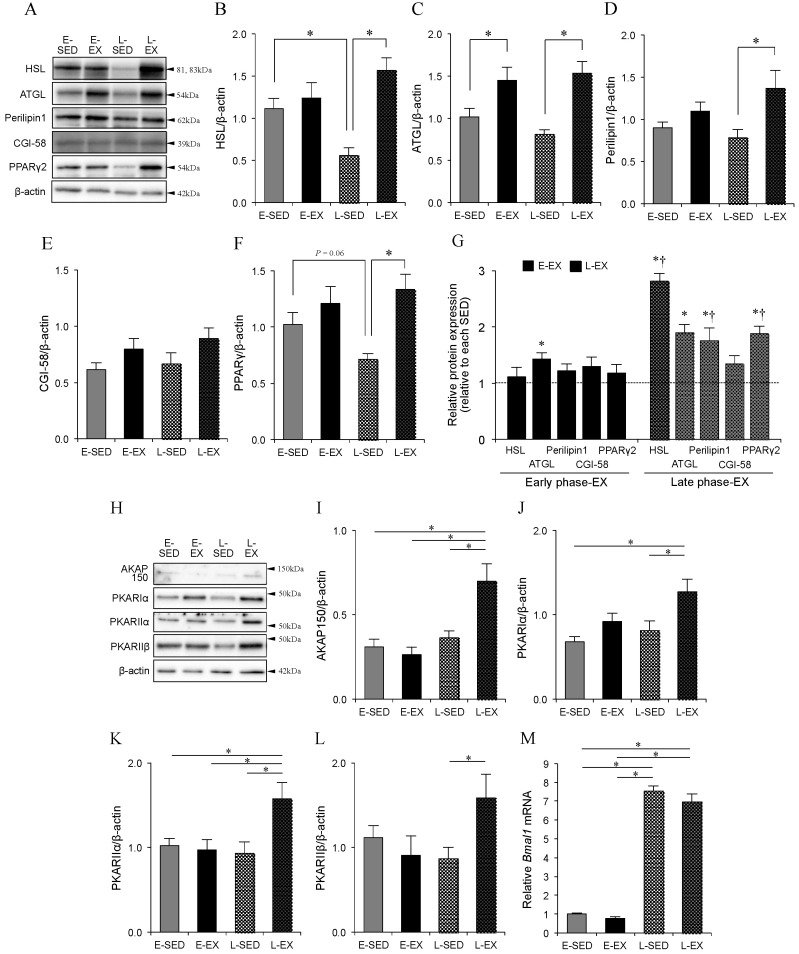
Effect of exercise training timing on the levels of lipolysis-associated proteins in primary adipocytes isolated from epididymal adipose tissues. (**A**) Representative immunoblotting data and the relative amounts of (**B**) HSL, (**C**) ATGL, (**D**) perilipin1, (**E**) CGI-58, and (**F**) PPARγ2 protein, in primary isolated adipocytes from rats, measured by Western blotting and normalized to the amount of β-actin. (**G**) Relative protein expression levels of each band in the E-EX and L-EX groups (levels in respective sedentary controls were set to 1). (**H**) Representative immunoblotting data and the relative levels of (**I**) AKAP150, (**J**) PKA-RIα, (**K**) PKA-RIIα, and (**L**) PKA-RIIβ protein measured by Western blotting and normalized to the level of β-actin. (**M**) *Bmal1* mRNA expression level measured by quantitative real-time PCR and normalized to the level of 18S rRNA. In all experiments, data are presented as the means ± S.E. (*n* = 6). * *p* < 0.05, SED group vs. EX group. ^†^
*p* < 0.05, E-EX vs. L-EX. The abbreviations for the groups of rats are shown in Figure 2 and Figure 3.

**Figure 6 ijms-21-06920-f006:**
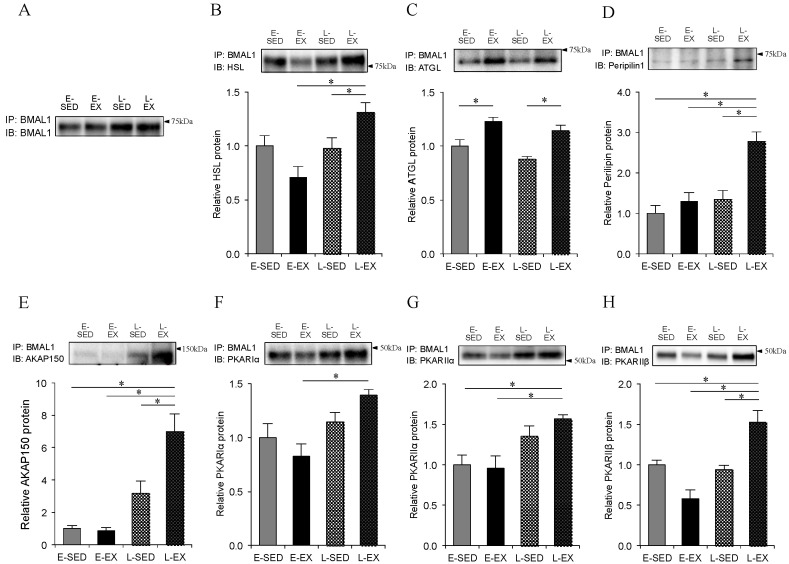
Association of BMAL1 with several lipolytic machineries in primary adipocytes isolated from epididymal adipose tissues. (**A**) Direct interaction between BMAL1 and BMAL1 revealed by the co-immunoprecipitation assay. Both lysates and immunoprecipitates (IP) were subjected to immunoblotting with (**B**) anti-HSL, (**C**) anti-ATGL, (**D**) anti-Perilipin1, (**E**) anti-AKAP150, (**F**) anti-PKA-RIα, (**G**) anti-PKA-RIIα, and (**H**) anti-PKA-RIIβ antibodies. Relative protein expression levels of each band in E-EX and L-EX groups (levels in respective sedentary controls are set to 1). In all experiments, data are presented as the means ± S.E. (*n* = 6). * *p* < 0.05. The abbreviations for the groups of rats are shown in Figure 2 and Figure 3.

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
