# Peer review of "Exercise Training-Enhanced Lipolytic Potency to Catecholamine Depends on the Time of the Day"

_ijms, 2020, doi:10.3390/ijms21186920_

Round 1
Reviewer 1 Report
The authors investigate here the phenomenon of exercise/catecholamine induced lipolysis in vitro and in rats. For this purpose, they first established an in vitro system consisting of synchronized, differentiated NIH3T3-L1 adipocyte cells. They showed a time-of-day dependent release of glycerol from the cultures, which was dependent on the circadian key regulator BMAL1. Then they established an in vivo model to analyze the corresponding time points in rats with and without exercise. Because they observed differences in isoproterenol-induced lipolysis, they further analyzed the lipolytic machinery. They found an increase of the lipase ATGL in response to exercise and further changes of the abundance of the lipolytic machinery and their regulatory components. Interestingly, BMAL1 associated with these complexes, although the function of this interaction remains unknown. As conclusion, the authors followed that exercise has a time-dependent effect on lipolysis in rats.
I guess this is an interesting study trying to link physiological data with its underlying molecular machinery. However, I have some concern regarding the data and their interpretation.
Major points
- In vitro data, Fig.1 and Fig.2: to rule out an effect of the signal transduction machinery, I guess it is necessary to perform a dose-response of the concentration of isoproterenol (30 um, 10 um, 3 um, and 1 um). Then it is possible to state that there are differences in the lipolytic machinery and that there is not simply changed the sensitivity of the response to isoproterenol.
- In vivo data, Fig.4 and Fig.5: essentially very nice data but the overall release of glycerol is not different between the two exercise groups (Fig.4A). Furthermore, the only component which fits the increase in glycerol release in response to exercise is the glycerol-releasing lipase ATGL (Fig.5C) and its interaction with BMAL1 (Fig.6C), even if the overall amount of Bmal1 mRNA (Fig.5H) and BMAL1 protein (Fig.6A) do not correlate with the regulatory phenotype. Hence, I don’t think that it is possible to state that the boosting mechanisms at the two different time points are different.
Minor points
- When you refer to “locus”, you probably mean complex?
- Abstract, line 31 insert protein kinase A (PKA)
- Change “18S mRNA” to 18S rRNA
- Line 185: you cannot state that the oscillation is independent of the SCN, if you do not ablate this brain center.
- Please clarify the paragraph lines 65-73. It’s written in a way that the two papers 25 and 26 sound very similar and I don’t see the point that there are time-of-day dependent differences.
Author Response
[Reviewer 1]
The authors investigate here the phenomenon of exercise/catecholamine induced lipolysis in vitro and in rats. For this purpose, they first established an in vitro system consisting of synchronized, differentiated NIH3T3-L1 adipocyte cells. They showed a time-of-day dependent release of glycerol from the cultures, which was dependent on the circadian key regulator BMAL1. Then they established an in vivo model to analyze the corresponding time points in rats with and without exercise. Because they observed differences in isoproterenol-induced lipolysis, they further analyzed the lipolytic machinery. They found an increase of the lipase ATGL in response to exercise and further changes of the abundance of the lipolytic machinery and their regulatory components. Interestingly, BMAL1 associated with these complexes, although the function of this interaction remains unknown. As conclusion, the authors followed that exercise has a time-dependent effect on lipolysis in rats.
I guess this is an interesting study trying to link physiological data with its underlying molecular machinery. However, I have some concern regarding the data and their interpretation.
Our reply to reviewer 1: Thank you for kind and valuable comments. We have revised our manuscript according your constructive advice and have highlighted the changes to our manuscript using blue coloured text. In accordance with this revision, some new references were cited and the consecutive numbers of the references were changed.
Major points:
In vitro data, Fig.1 and Fig.2: to rule out an effect of the signal transduction machinery, I guess it is necessary to perform a dose-response of the concentration of isoproterenol (30 μM, 10 μM, 3 μM, and 1 μM). Then it is possible to state that there are differences in the lipolytic machinery and that there is not simply changed the sensitivity of the response to isoproterenol.
→ Thank you for your constructive suggestion. As per your suggestion, the data for the dose-dependent effect of isoproterenol on adipocyte lipolysis may strongly help understand the underlying mechanism. In the current study, the isoproterenol dose was based on previous studies on 3T3-L1 adipocytes [Ref. 55; Naunyn Schmiedebergs Arch Pharmacol 1999, 359:310-321, Ref. 56; Br J Pharmacol, 1997, 120: 201-210]; these studies demonstrated the dose-dependent effect of isoproterenol on lipolysis in 3T3-L1 adipocytes, and found the maximally effective concentration of isoproterenol to be 10 μM. We have included this description in the materials and methods section using blue coloured text (Lines 556-558). In this context, we have added two citations (Refs. 55 and 56).
In vivo data, Fig.4 and Fig.5: essentially very nice data but the overall release of glycerol is not different between the two exercise groups (Fig.4A). Furthermore, the only component which fits the increase in glycerol release in response to exercise is the glycerol-releasing lipase ATGL (Fig.5C) and its interaction with BMAL1 (Fig.6C), even if the overall amount of Bmal1 mRNA (Fig.5H) and BMAL1 protein (Fig.6A) do not correlate with the regulatory phenotype. Hence, I don’t think that it is possible to state that the boosting mechanisms at the two different time points are different.
→ Thank you for your constructive suggestion. Although the overall glycerol release was not different between the E-EX and L-EX groups (Fig. 4A), isoproterenol-stimulated lipolysis expressed as fold induction over basal lipolysis was significantly greater in the L-EX group than in the E-EX group (Fig. 4B). Further, the level of Ser660-phosphorylated HSL was significantly higher in the L-EX group than in the E-EX group (Fig. 4F). For further confirmation, we conducted additional experiments and found that even though the L-EX rats exhibited greater protein expression only for AKAP150 and the regulatory subunits PKA-RIIα compared to the E-EX rats, high associations of BMAL1 were found with the lipolytic machineries tested except for ATGL. Together, we consider that the greater lipolytic potency of adipocytes to isoproterenol in L-EX rats compared to E-EX rats could be attributed to their high interaction of BMAL1 with the AKAP-PKA-HSL-Perilipin1 locus. Thus, we added new figures (Fig. 5H-L and Fig. 6D) and added a new interpretation in the Discussion section using blue coloured text (Lines 364-375, in the revised manuscript). We have also cited two additional references [Ref. 40; J Biol Chem, 2006, 281, 15837-15844, Ref. 41; EMBO J, 2011, 30, 4371-4386].
Minor points:
When you refer to “locus”, you probably mean complex?
→ Thank you for your query. For clarification, we have added the following explanation to the text: AKAP-PKA-HSL-Perilipin1, the locus for essential events in the sequential activation of triacylglycerol hydrolysis (lines 374-375 in the revised manuscript).
Abstract, line 31 insert protein kinase A (PKA) Change “18S mRNA” to 18S rRNA
→ Thank you for your constructive suggestion. We have made the required change.
Line 185: you cannot state that the oscillation is independent of the SCN, if you do not ablate this brain center.
→ Thank you for your constructive comment. Accordingly, because we did not ablate the SCN, we have rewritten lines 161-162 in the revised manuscript as follows: These results indicate that the circadian clockwork can oscillate accurately in adipose tissue, similar to that described in previous studies [Refs. 22-27].
Please clarify the paragraph lines 65-73. It’s written in a way that the two papers 25 and 26 sound very similar and I don’t see the point that there are time-of-day dependent differences.
→ Thank you for your constructive suggestion. According to both your comment and the third reviewer’s comment, we have deleted this paragraph. However, because the findings of these studies [Ref. 45; Sato et al., Cell Metab 2019, Ref. 46; Ezagouri et al., Cell Metab 2019] are very beneficial to understand our findings, we have tried to discuss this point in lines 410-415 of the revised manuscript as follows: Our present results might also be associated with the recent finding that exercise in the late dark phase ameliorates diet-induced obesity by affecting energy homeostasis rather than glucose homeostasis [Refs. 46 and 47], and through enhanced metabolic activity in the skeletal muscles [Refs. 48 and 49]. The energy metabolism of skeletal muscles during acute exercise has also been shown to differ greatly depending on the time of the day for a single bout of exercise [Refs. 48 and 49].
Reviewer 2 Report
Kato and colleagues present a finding that adipose lipolysis is dependent on adipocyte clock proteins, notably BMAL1, that interact with HSL and ATGL. This results in lipolysis being dependent on exercise and time of day in rats and cells. In total, it is in an interesting finding. There are some questions raised regarding how the data is collected and or expressed relative to the conclusions drawn. Some of these are currently discussed by the authors, while others could be corrected with a different data presentation and, most certainly discussed. A few minor points then follow.
The trained animal body weights are a concern relative to their sedentary peers. In may ways this result is expected. The authors address how this impacts drug dosing, but it would also have impacts on other factors that will influence local protein expression/activation and glycerol appearance: insulin levels, lipogenic v. lipolytic machinery, vascular tone, adrenergic activity and substrate availability (e.g., glucose levels and glycolysis versus FAO in the adipose itself and systemically).
Isoproterenol is necessary in the tissue culture experiments, but a beta3 agonist could be used in vivo... would the authors expect the same result with a central versus local lipolytic stimulant? What are insulin levels in the rats as a repressive signal?
How much of the result in the immunoprecipitation experiments are merely the result of higher protein expression? Blotting for the pulled down protein would help to demonstrate this on each blot (not just a "representative" one).
Glycerol is graphed as a timepoint? Following lipolysis, serum glycerol levels will increase over time and then recede over hours allowing for an AUC calculation of the total released. A spot check in an animal does not provide sufficient information. The result will also be challenging to judge due to weights.
Figure 2: westerns are not labeled, C/D have 4 groups graphed but the WB shows only 2 groups.
Minor:
The discussion is largely a rehash of the results and should not be referencing specific results by Figure panel. Please focus on the context of the conclusions in the field.
Line 65: while unfortunate to hear, it is not necessary for the authors to point this out.
Line 76: need a comma after 'basal lipolysis' to close that clause.
Final intro para: too much results in here, rather the hypothesis and plan should be written here
Author Response
[Reviewer 2]
Kato and colleagues present a finding that adipose lipolysis is dependent on adipocyte clock proteins, notably BMAL1, that interact with HSL and ATGL. This results in lipolysis being dependent on exercise and time of day in rats and cells. In total, it is in an interesting finding. There are some questions raised regarding how the data is collected and or expressed relative to the conclusions drawn. Some of these are currently discussed by the authors, while others could be corrected with a different data presentation and, most certainly discussed. A few minor points then follow. The trained animal body weights are a concern relative to their sedentary peers. In may ways this result is expected. The authors address how this impacts drug dosing, but it would also have impacts on other factors that will influence local protein expression/activation and glycerol appearance: insulin levels, lipogenic v. lipolytic machinery, vascular tone, adrenergic activity and substrate availability (e.g., glucose levels and glycolysis versus FAO in the adipose itself and systemically).
Our reply to reviewer 2: Thank you for kind and valuable comments. We have revised our manuscript according your constructive advice and have highlighted the changes to our manuscript in blue coloured text.
We strongly agree to your comment. Therefore, we have described this point as follows (line 399-402 in the revised manuscript): Further, the possible effects of diurnal variations including glucose and lipid metabolism [Ref. 42], and of hormones such as cortisol and insulin that affect the lipolytic response [Refs. 43-46], should be considered more importantly. Finally, other factors that impact local protein expression/activation in adipocytes and that determine blood glycerol concentration such as insulin levels, adipogenic versus lipolytic mechanisms, vascular tone, adrenergic activity, and substrate availability, should also be considered more systematically.
Isoproterenol is necessary in the tissue culture experiments, but a beta3 agonist could be used in vivo... would the authors expect the same result with a central versus local lipolytic stimulant? What are insulin levels in the rats as a repressive signal?
→ Thank you for your constructive suggestion. As per your comment, it is important to consider whether the β3-receptor specific agonist could be used to verify lipolysis in vivo. However, noradrenaline elicits a marked lipolytic effect with an intrinsic activity equivalent to that of isoproterenol, but always with a lower potency than that of isoprenaline [Ref. 55: Carpene C, et al. Naunyn-Schmiedeberg’s Arch Pharmacol,1999, 359: 310–321], and the maximal lipolytic responsiveness of adipocytes is not significantly different between noradrenaline and isoproterenol [Ref. 56: Germack R, et al., Br J Pharmacol, 1997, 120: 201-210]. Thus, we expect the same result with a central versus local maximal lipolytic stimulation.
The lipolytic response is well known to be affected by insulin levels. In this context, we confirmed that the insulin levels did not differ between the E-EX group and the L-EX group (data not shown). However, because the enhanced antilipolytic action of insulin is reported to be significantly greater in the trained than in the control group (Suda K, et al., J Appl Physiol, 1993, 74: 2935-2939), future studies are required to verify whether the antilipolytic action of insulin could be modified by the timing of exercise.
How much of the result in the immunoprecipitation experiments are merely the result of higher protein expression? Blotting for the pulled down protein would help to demonstrate this on each blot (not just a "representative" one).
→ Thank you for your valuable and constructive suggestion. To respond to your comment with evidence, we conducted additional experiments. We found that although greater protein expression levels in the L-EX group compared to the E-EX group were found only for AKAP150 and the regulatory subunits of PKA-RIIα (Fig. 5H-L), the immunoprecipitates of HSL, Perilipin1, AKAP150, and PKA-RIIβ were significantly higher in in the L-EX group compared to the E-EX group (Fig. 6 B, D, E-H). Thus, the increase of the immunocomplex in the L-EX group cannot be explained only by the increase in the protein expression level of the lipolytic machineries in the L-EX group compared to the E-EX group. This point is described in lines 364-375 of the revised manuscript.
Glycerol is graphed as a timepoint? Following lipolysis, serum glycerol levels will increase over time and then recede over hours allowing for an AUC calculation of the total released. A spot check in an animal does not provide sufficient information. The result will also be challenging to judge due to weights.
→ Thank you for your constructive suggestion. As per your comment, glycerol release was graphed as a timepoint. The current in vitro experimental design was based on our previous studies in isolated primary adipocytes from exercise-trained rats [Refs. 6 and 10]. We strongly agree that calculating the AUC of total glycerol release could be valuable. However, because training experiments take a lot of time, we hope to make it a future research topic.
Figure 2: westerns are not labeled, C/D have 4 groups graphed but the WB shows only 2 groups.
→ Thank you for your constructive suggestion. According to your suggestion, we have corrected the graph label in figures 2C and D.
Minor points:
The discussion is largely a rehash of the results and should not be referencing specific results by Figure panel. Please focus on the context of the conclusions in the field.
→ Thank you for this constructive suggestion. In accordance with the reviewer’s comment, we have rewritten the discussion part.
Line 65: while unfortunate to hear, it is not necessary for the authors to point this out.
→ Thank you for your constructive suggestion. According to your comment and that by reviewer 1, we decided to delete this paragraph. However, because the findings of these studies [Ref. 48; Sato et al., Cell Metab 2019, Ref. 49; Ezagouri et al., Cell Metab 2019] are very beneficial to understand our findings, we have inserted the following description in lines 410-415 of the revised manuscript: Our present results might also be associated with the recent finding that exercise in the late dark phase ameliorates diet-induced obesity by affecting energy homeostasis rather than glucose homeostasis [Refs. 46 and 47], and through enhanced metabolic activity in the skeletal muscles [Refs. 48 and 49]. The energy metabolism of skeletal muscles during acute exercise has also been shown to differ greatly depending on the time of the day for a single bout of exercise [Refs. 48 and 49].
Line 76: need a comma after 'basal lipolysis' to close that clause.
→ Thank you for your constructive suggestion. Because the final paragraph of the introduction in the original manuscript was deleted in the revised manuscript, we did not address this point.
Final intro para: too much results in here, rather the hypothesis and plan should be written here.
→ Thank you for your constructive suggestion. According to your advice, we tried to rewrite the introduction focusing on the hypothesis and plan in lines 64-66 of the revised manuscript.
Reviewer 3 Report
In the current manuscript, Kato et al set to test whether if there is a relationship between the timing of exercise training and its effect on adipocyte lipolysis. The authors do a thorough description of the recent findings in their field of work, acknowledging current publications and highlighting what is novel about their work. I believe this work is valuable and therefore worth to be published if they are able to address of the comments that are mentioned bellow.
1-The author might consider splitting the figure 1A into 2 graphs so the scale can be adjusted to show better Bmal1 oscillatory behavior. Since HSL phosphorylation seems to respond to a circadian pattern, did the authors consider analyzing the kinase involved in the phosphorylation of these sites in vitro? (e.g. PKA levels).
2-On figure 2 the authors show that Bmal1-KD reduced HSL mRNA and protein levels (2C-D) but this is not reproduce on figure 2F, where the decrease in HSL protein levels is not as marked as before.
3-On Figure 5A L-SED seems to have lower overall loading (even the actin seems a bit lower) I would advice the authors to re-run this blot.
4-On figure 6, could the higher association in AKAP150 be due to a higher abundance of the protein? To answer that the authors need to show total AKAP150 and PKA regulatory subunits levels.
Author Response
[Reviewer 3]
In the current manuscript, Kato et al set to test whether if there is a relationship between the timing of exercise training and its effect on adipocyte lipolysis. The authors do a thorough description of the recent findings in their field of work, acknowledging current publications and highlighting what is novel about their work. I believe this work is valuable and therefore worth to be published if they are able to address of the comments that are mentioned bellow.
Our reply to reviewer 3: Thank you for your kind and valuable comments. We have revise our manuscript according your constructive advice and have highlighted the changes to our manuscript using blue coloured text.
1-The author might consider splitting the figure 1A into 2 graphs so the scale can be adjusted to show better Bmal1 oscillatory behavior. Since HSL phosphorylation seems to respond to a circadian pattern, did the authors consider analyzing the kinase involved in the phosphorylation of these sites in vitro? (e.g. PKA levels).
→ Thank you for your constructive suggestion. Although we did not split figure 1A into 2 graphs, we have adjusted the scale in figure 1A to distinguish between Bmal1 and Per2 mRNA oscillatory behaviors more clearly. In addition, we also tested the expression of Peripilin, AKAP150 and the regulatory subunits (RIα, RIIα, and RIIβ) of PKA (Fig. 2D-G) in the remaining samples of 3T3-L1 adipocytes with BMAL1 knockdown, and have added the results to the revised paper (lines 113-121, in the revised manuscript).
2-On figure 2 the authors show that Bmal1-KD reduced HSL mRNA and protein levels (2C-D) but this is not reproduce on figure 2F, where the decrease in HSL protein levels is not as marked as before.
→ Thank you for this constructive suggestion. According to your advice, we have re-run the HSL protein in Figure 2F, and have presented new blots.
3-On Figure 5A L-SED seems to have lower overall loading (even the actin seems a bit lower) I would advice the authors to re-run this blot.
→ Thank you for your constructive suggestion. Accordingly, we determined the abundance of β-actin by re-running the blot of adipocyte lysates with the same loading volume in different groups. We believe that you can confirm the original image submitted as a supplementary file for reviewers only.
4-On figure 6, could the higher association in AKAP150 be due to a higher abundance of the protein? To answer that the authors need to show total AKAP150 and PKA regulatory subunits levels.
→ Thank you for your valuable and constructive suggestion. Accordingly, we conducted additional experiments and found that although greater expression levels of the proteins tested in the L-EX group compared to the E-EX group were found in only for AKAP150 and the regulatory subunits of PKA-RIIα (Fig. 5H-L), the immuprecipitates of HSL, Perilipin1, AKAP150, and PKA-RIIβ were significantly higher in the L-EX group compared to the E-EX group (Fig. 6 B, D, E-H). Thus, the increase in the immunocomplex in the L-EX group cannot be explained by the increase in the protein expression level of lipolytic machineries themselves in the L-EX group compared to the E-EX group. This context is described in lines 364-375 of the revised manuscript.

Round 2
Reviewer 2 Report
The authors have addressed most of the concerns that were raised previously. What remains is minor, but significant. From before:
"The discussion is largely a rehash of the results and should not be referencing specific results by Figure panel. Please focus on the context of the conclusions in the field.
→ Thank you for this constructive suggestion. In accordance with the reviewer’s comment, we have rewritten the discussion part."
Despite their comment, the authors have not addressed this as there are still direct references to figures (ex, line 312 and 328 and more) and data. Some minor editing would eliminate these.
Author Response
The authors have addressed most of the concerns that were raised previously. What remains is minor, but significant. From before:
"The discussion is largely a rehash of the results and should not be referencing specific results by Figure panel. Please focus on the context of the conclusions in the field.
→ Thank you for this constructive suggestion. In accordance with the reviewer’s comment, we have rewritten the discussion part."
Despite their comment, the authors have not addressed this as there are still direct references to figures (ex, line 312 and 328 and more) and data. Some minor editing would eliminate these.
→ We are sorry we didn’t make it clear enough. According to your constructive comment, we have retouched the discussion by deleting the referencing specific results. We sincerely agree with your suggestion that the discussion should be focused on the context the conclusions in the field. We would like to make use of your valuable advice in our future manuscripts.
Reviewer 3 Report
The authors addressed all the comments so I’m pleased to accept the manuscript in its current form.
Author Response
The authors addressed all the comments so I’m pleased to accept the manuscript in its current form.
→ We wish to thank the reviewer for this comment. Both, the original and revised manuscripts has been edited by a professional English language editing service (Editage). The certificate for English editing is being submitted with this revised manuscript.
